# Estimating the impact of the 1991 Pinatubo eruption on mesospheric temperature by analyzing HALOE (UARS) temperature data

Sandra Wallis, Christoph Gregor Hoffmann, and Christian von Savigny

Institute of Physics, University of Greifswald, Greifswald, Germany

**Correspondence:** Sandra Wallis (sandra.wallis@uni-greifswald.de)

**Abstract.** The Mt. Pinatubo eruption in 1991 had a severe impact on the Earth system with a well-documented warming of the tropical lower stratosphere and a general cooling of the surface. This study focuses on the impact of this event on the mesosphere by analyzing solar occultation temperature data from the Halogen Occultation Experiment (HALOE) instrument on the Upper Atmosphere Research Satellite (UARS). Previous analysis of lidar temperature data found positive temperature anomalies of up to 12.9 K in the upper mesosphere that peaked in 1993 and were attributed to the Pinatubo eruption. Fitting the HALOE data according to a previously published method indicates a maximum warming of the mesosphere region of 4.1 ± 1.4 K and does not confirm significantly higher values reported for that lidar time series. An alternative fit is proposed that assumes a more rapid response of the mesosphere to the volcanic event and approximates the signature of the Pinatubo with an exponential decay function having an e-folding time of 6 months. It suggests a maximum warming of 5.4 ± 3.0 K if the mesospheric perturbation is assumed to reach its peak 4 months after the eruption. We conclude that the HALOE time series probably captures the decay of a Pinatubo-induced mesospheric warming at the beginning of its measurement period.

## 1 Introduction

Explosive volcanic eruptions inject large amounts of gases and ash into the atmosphere with a potential impact on global climate. $SO_2$ that reaches the stratosphere will be converted to sulfate aerosols that remain in the stratosphere for up to several years (Barnes and Hofmann, 1997). These aerosols alter the net radiative balance and lead to a warming of the stratosphere and a cooling of the troposphere (Robock, 2000).

The Pinatubo eruption in June 1991 was one of the most explosive eruptions in the last centuries and introduced approximately 10 to 20 Mt of $SO_2$ into the stratosphere. The great amount of sulfate aerosols increased the stratospheric temperature up to 3.5 K (Kilian et al., 2020) and cooled the surface globally by 0.5 K (Parker et al., 1996). The impact of volcanic eruptions on the upper part of the middle atmosphere is less investigated (von Savigny et al., 2020). It is established that the atmospheric layers are not isolated but interact with each other. Dynamic interactions between the troposphere and the mesosphere via gravity waves can transport energy vertically and have a significant impact on the temperature field of the middle atmosphere. A model simulation introducing volcanic aerosols with an optical depth of 0.15 into the lower stratosphere showed an increase in mesospheric temperature due to dynamics (Rind et al., 1992). Analysis of Na lidar temperature data by She et al. (1998) at northern mid-latitudes (40.6° N) showed a strong episodic warming with a maximum increase of 9 K at 86 km and 12.9 K at

100 km in 1993. This was attributed to the Pinatubo eruption in June 1991. Kalicinsky et al. (2016) reported annual averaged OH(3-1) rotational temperature above Wuppertal (51.3° N) with a strong peak in 1991. They added a vertical line marking the Mt. Pinatubo eruption to their graph but did not discuss a connection between the Pinatubo eruption and the temperature signature further in their paper. Offermann et al. (2010) also noticed a temperature maximum in 1991 after de-trending the OH(3-1) Wuppertal temperature time series, which they connected with the eruptions of Mt. Pinatubo and Cerro Hudson. Analysing Rayleigh lidar data, Keckhut et al. (1995) observed a significant warming of 5 K in the mesosphere from 60 – 80 km in the summer of 1992 and 1993 and at 44° N. They associated this finding with the Pinatubo eruption. Similarly, annually averaged temperature data from the High Resolution Doppler Imager on the Upper Atmosphere Research Satellite (UARS) exhibited a 5 K warming at 100 km in the years 1992 and 1993 compared to the observed time period of 1992 – 1998 which might be related to the Pinatubo eruption according to the authors (Thulasiraman and Nee, 2002).

In a recent modelling study Ramesh et al. (2020) employed the Whole Atmosphere Community Climate Model version 6 (WACCM6) model to investigate the long-term variability of middle atmospheric temperature and zonal wind caused by different drivers, including volcanic perturbations of the stratospheric aerosol optical depth. The simulations showed volcanic perturbations to the atmospheric temperature field reaching up into the lower thermosphere. However, these responses were not significant in many latitude-altitude regions and the underlying physico-chemical processes were not discussed in Ramesh et al. (2020).

Some researchers concluded that no episodic perturbation from the Pinatubo eruption is evident in their mesospheric temperature data. Remsberg (2009) did not include an episodic forcing for the analysis of HALOE temperature time series at altitudes up to 80 km starting in 1991. Remsberg and Deaver (2005) acknowledged the interpretation by She et al. (1998) of the temperature increase as being caused by Pinatubo although they did not consider a volcanic contribution for their analysis and only reported a maximum of an 11 year solar cycle-like term around 1993 (Remsberg et al., 2002b). Bittner et al. (2002) reported OH(3-1) rotational temperatures in 87 km altitude at 51.3° N and fitted one year of data with a regression, moved the analysis window by half a year and kept repeating the procedure. They found a clear perturbation in the phases of the seasonal fits in the middle of 1991 that vanished half a year later. A correlation between the volcanic eruption and the temperature signal, however, was considered inconclusive. They could not reproduce the findings from She et al. (1998) and Keckhut et al. (1995), but speculated that this discrepancy could be due to the location of the ground-based measurements. According to the authors, the mountains close to the measurement site of Fort Collins used in the study of She et al. (1998) could enhance gravity wave formation – a possible route for energy transfer from the stratosphere to the mesosphere.

There are very few data sets of the middle atmospheric temperature covering the period around or after the eruption of Mt. Pinatubo. One of these data sets was provided by the Halogen Occultation Experiment (HALOE) that was launched on the Upper-Atmosphere Research Satellite (UARS) and started its scientific observations in October 1991. We use the HALOE temperature data to investigate if the mesosphere was perturbed by the explosive 1991 Mt. Pinatubo eruption and give an estimate of its potential impact.

This paper is structured as follows. In section 2 we describe the HALOE temperature data set and the regression approaches chosen in this study to model the temperature data. Section 3 describes the main results of the regression analysis, followed by a discussion of the implications of the detected temperature signals in section 4. Conclusions are presented in section 5.

## 2   Data and data analysis

The impact of the atmospheric perturbation by the Pinatubo eruption on mesospheric temperature is estimated in this work by using temperature data from the Halogen Occultation Experiment (HALOE) (Russell III et al., 1993) on board the Upper Atmosphere Research Satellite (UARS) (Reber et al., 1993). HALOE started its scientific observations using solar occultation on October 11, 1991 (Russell III et al., 1993) and should be able to track a potential remaining signature from the Pinatubo eruption. The NASA temperature data product was retrieved from the $CO_2$ transmission in the altitude range between approximately 35 and 85 km (e.g., Hervig et al., 1996; Remsberg, 2009). HALOE took about 15 sunrise and 15 sunset measurements every day. Due to the non-sun-synchronous orbit of the UARS spacecraft, the latitudes of HALOE occultation measurements changed slowly from day to day. Latitudes between 50° N and 50° S are typically covered in each month. Depending on the UARS yaw cycle, latitudes up to 80° are covered in one hemisphere.

In this study we used HALOE level 2 (Version 19) temperature data from the NASA Goddard Earth Sciences Data and Information Services Center website (https://disc.gsfc.nasa.gov) and the F10.7cm solar flux provided by the Laboratory for Atmospheric and Space Physics Interactive Solar Irradiance Data Center (https://lasp.colorado.edu/lisird/data/penticton_radio_flux/) as a solar proxy.

The HALOE files contain the retrieved temperature at 0.3 km spacing, the latitudes and the classification of the measurement as sunset and sunrise measurements, respectively (Thompson and Gordley, 2009). Only temperatures between 43 and 87 km altitudes and with values not equal to 0 were used. Measurements that were tagged as being problematic due to their corresponding trip or lockdown angle (http://haloe.gats-inc.com/user_docs/index.php) were dismissed from the analysis. All temperature data was divided into sunset and sunrise measurements and each data set was zonally and daily averaged and binned in 10° latitude bins if at least two measurements existed for this time/latitude bin. Afterwards, the mean values of the sunset and sunrise data set for each latitude and altitude bin, $mean_{ss}$ and $mean_{sr}$, were calculated to correct the data according to equation 1 which is adapted from Remsberg et al. (2002b).

$$T_{ss,corr} = T_{ss} - (mean_{ss} - mean_{sr})/2.$$
$$T_{sr,corr} = T_{sr} + (mean_{ss} - mean_{sr})/2. \tag{1}$$

The corrected sunset and sunrise data $T_{ss,corr}$ and $T_{sr,corr}$ were combined to a single data set. If both sunset and sunrise data exist for the same latitude-altitude bin, the average of both values was taken. Subsequently, the monthly average was obtained if at least two measurements existed for this bin and month.

If not stated otherwise, a fit was applied including a constant and linear term, seasonal terms, a solar proxy and an episodic perturbation term. The seasonal terms account for the annual ($P_{year}$), semi-annual ($P_{semiyear}$), 4-months ($P_{4months}$) and 3-

90   months ($P_{3months}$) oscillations while the F10.7cm solar proxy was used to include the solar contribution ($S_{F10.7cm}$) term. Two different approaches are proposed and compared. The first approach adopts an expression for the episodic perturbation term from She et al. (1998) which is also considered as the last term in equation 2

$$
\begin{aligned}
F_1(t) = & A_0 + A_1 \cdot t + A_3 \cdot cos(2\pi \cdot \frac{t - A_2}{P_{year}}) \\
& + A_4 \cdot cos(2\pi \cdot \frac{t - A_5}{P_{semiyear}}) \\
& + A_6 \cdot cos(2\pi \cdot \frac{t - A_7}{P_{4months}}) \\
& + A_8 \cdot cos(2\pi \cdot \frac{t - A_9}{P_{3months}}) \\
& + A_{10} \cdot S_{F10.7cm}(t) \\
& + A_{11} \cdot \frac{2}{exp(\frac{-(t-t_0)}{t_1}) + exp(\frac{t-t_0}{t_2})}
\end{aligned}
\tag{2}
$$

with $P_{year}$= 1 year, $P_{semiyear}$= 0.5 years, $P_{4months}$=0.33 years and $P_{3months}$= 0.25 years. The parameters $t_0$, $t_1$ and $t_2$ indicate
the delay, rise and decay time (She et al., 2015) and were varied as described in the text below.

During the analysis of the HALOE temperature data it became apparent that for some altitude-latitude bins there appeared to be an anomalous enhancement of the temperature values right at the beginning of the time series in October 1991. This temperature anomaly was found to decay roughly exponentially. This finding motivated the use of an alternative fitting approach based on two consecutive fit functions. After the constant, linear, seasonal and the solar contributions are accounted for in
equation 3, an exponential decay function is fitted to the residual from the first fit in order to capture a potential volcanic perturbation, as can be seen in equation 4.

$$
\begin{aligned}
F_2(t) = & B_0 + B_1 \cdot t + B_3 \cdot cos(2\pi \cdot \frac{t - B_2}{P_{year}}) \\
& + B_4 \cdot cos(2\pi \cdot \frac{t - B_5}{P_{semiyear}}) \\
& + B_6 \cdot cos(2\pi \cdot \frac{t - B_7}{P_{4months}}) \\
& + B_8 \cdot cos(2\pi \cdot \frac{t - B_9}{P_{3months}}) \\
& + B_{10} \cdot S_{F10.7cm}(t)
\end{aligned}
\tag{3}
$$

$$
F_3(t) = C_0 \cdot exp(\frac{-t}{0.5})
\tag{4}
$$

All fits were performed with equal weights and the regression coefficient $B_{10}$ of the solar proxy term was limited to positive
values, including zero. Confidence intervals for the regression coefficients were determined with a jackknife method. If the

confidence interval does not include zero than the regression coefficient is considered significant. This method is described in more detail in Appendix A1. The uncertainties for the amplitudes are reported as the standard errors of the jackknife estimate (Efron, 1981).

## 3 Results

This study analyses the HALOE temperature data for a potential Pinatubo-related episodic warming in the mesosphere. We fit the temperature data and compare two approaches that each include a term for a potential volcanic perturbation (for the results of the fit without any volcanic contribution, please see Figure S1 in the supplementary material and compare them with Figure 1 and Figure 2). In our first approach, we include a perturbation term in our fit routine that was suggested by a Na lidar study (She et al., 1998) that reported a strong temperature perturbation. We can now compare our results to their findings.

Figure 1 presents the HALOE temperature time series at $40 \pm 5°N$ and 86 km, similar to the location of the lidar station at 41°N and the altitude reported in the lidar study (She et al., 1998). Some difference to the result of the ground-based lidar stations due to the zonal averaging are, nevertheless, expected as will be discussed below. This study uses a data product where the HALOE measurements from 76.5 km to 99 km have been merged with the MSIS (Mass Spectrometer – Incoherent Scatter) climatology during the processing. According to Remsberg et al. (2002a) the HALOE temperatures up to 87.5 km altitude

can be assumed to be almost completely independent of MSIS and can thus be used for a direct comparison. The time series was fitted according to equation 2, containing a constant and a linear term, seasonal terms, a solar proxy and the episodic perturbation term. The parameters $t_0$ to $t_2$ that define the perturbation term are chosen according to the literature (She et al., 1998). The resulting fit is drawn as a green curve over the data. Figure 1 clearly shows that the fit successfully captures the measured data points. Figure 2 shows the residual determined by subtracting the fit from the actual data points. No modulation

or remaining signature is visible in the residual which further suggests that the fit is based on valid assumptions.

We now focus on the amplitude (maximum or minimum value) of the episodic perturbation term. The amplitude of the perturbation is negative (-3.5 $\pm$ 1.5 K) and indicates a cooling between 1992 – 1994 in contrast to the warming reported previously (She et al., 1998). Although the fit as a whole was performed successfully, the amplitude and sign of the term that captures the episodic perturbation contradict previous findings.

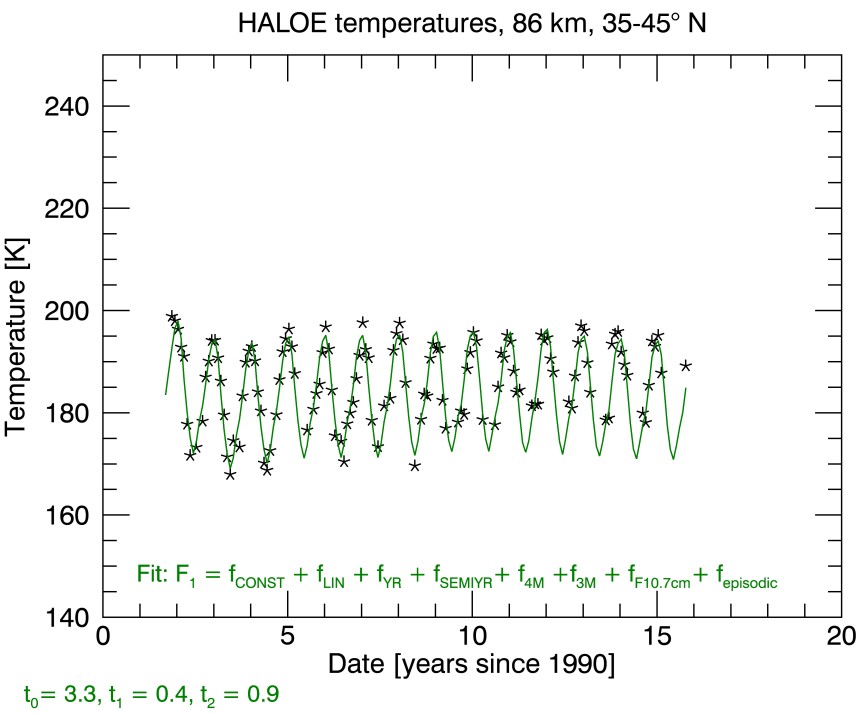

**Figure 1.** The HALOE temperatures series at 86 km and $40 \pm 5°$N is plotted together with the fit curve $F_1$ from equation 2. The curve successfully captures the data points indicating that the assumptions of the fit are valid.

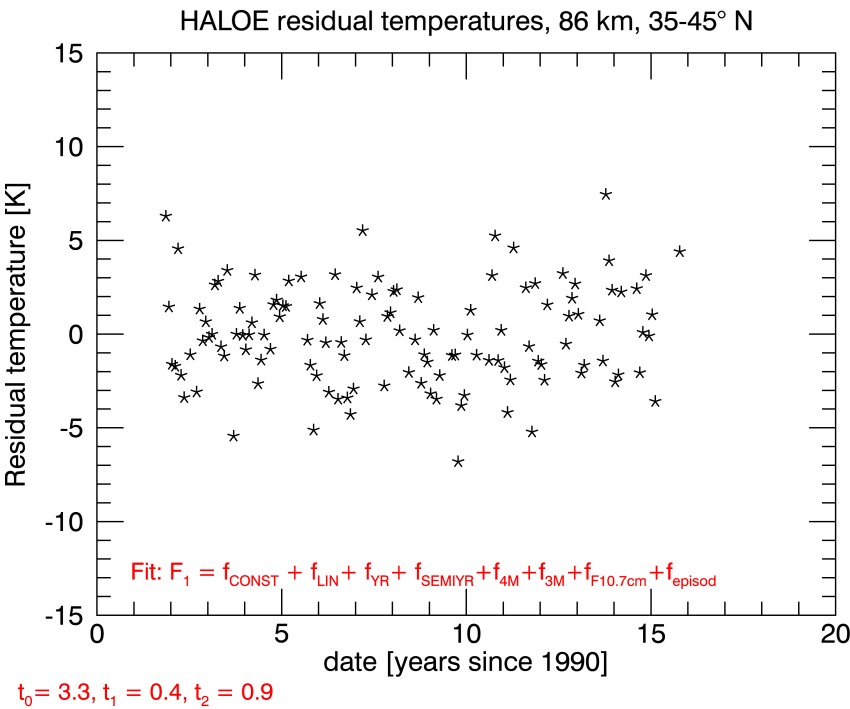

**Figure 2.** This graph demonstrates that no signature remains in the residual after the best estimate for fit $F_1$ was subtracted from the HALOE temperatures series at 86 km and $40 \pm 5°$ N.

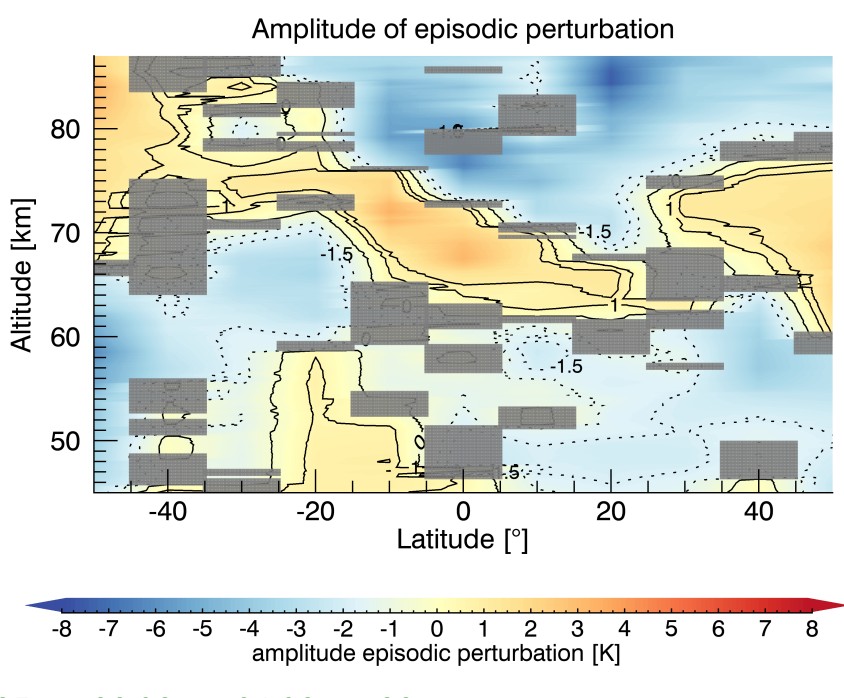

2.7 ≤$t_0$≤ 3.3, 0.2 ≤$t_1$≤ 0.4, 0.9 ≤$t_2$≤ 2.9

**Figure 3.** The amplitude of the episodic perturbation has its greatest positive values at 50°S in 84 km altitude but exhibits also high values at the equator and at an altitude between 65 – 70 km. The fit can vary the parameters $t_0$ to $t_2$ for the episodic perturbation in the following ranges: 2.7≤$t_0$≤3.3, 0.2≤$t_1$≤0.4 and 0.9≤$t_2$ ≤2.9. Grey areas indicate amplitudes that are not significant.

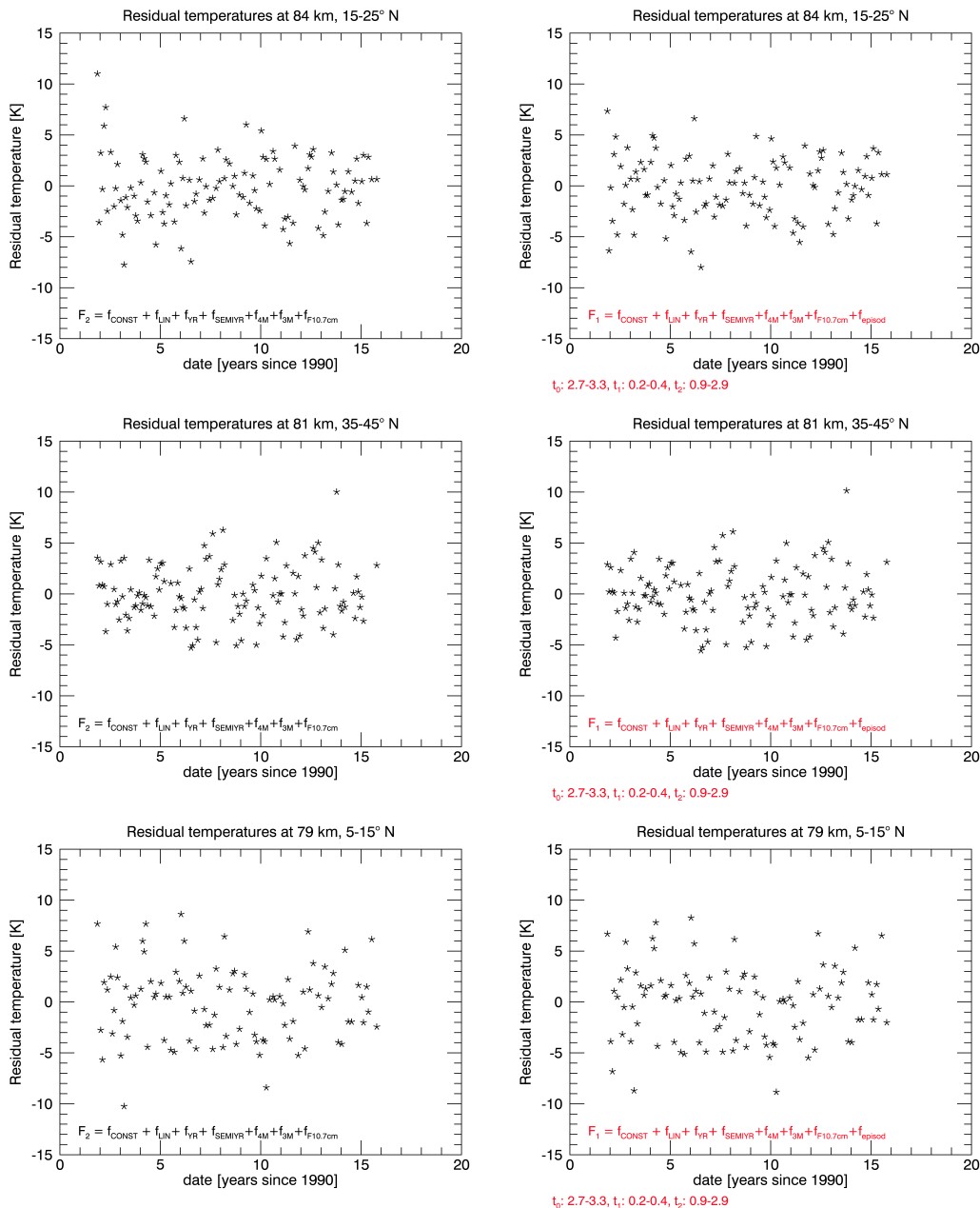

**Figure 4.** Comparison of fit residuals for three selected latitude-altitude combinations for fits without a volcanic perturbation term (left column) and with a volcanic perturbation term considered in the fit (right column), respectively. Some time series show a clear potential Pinatubo signal (top row) whereas no signature is visible with the naked eyes in others (example in the middle row). We further included a randomly selected pair (bottom).

The analysis was expanded to the middle atmosphere from 45 to 87 km altitude and from 50°S to 50°N. The variables $t_0$ to $t_2$ are now treated as fit parameters. In contrast to the regression coefficient $A_{11}$, they are only allowed to vary in defined ranges ($2.7 \leq t_0 \leq 3.3$, $0.2 \leq t_1 \leq 0.4$ and $0.9 \leq t_2 \leq 2.9$) on the basis of a previous lidar study (She et al., 1998). The amplitude of the episodic perturbation for the middle atmosphere is presented as a contour plot in Figure 3. Its magnitude and sign might hint to a Pinatubo-induced temperature signature in the atmosphere. A grey color indicates that this area is not significant (for

a detailed description, please see Appendix A1). The amplitude at 40°N and at 86 km slightly increases to -3.4 ± 1.3 K, but is still negative. The highest positive values of 4.1 ± 1.4 K occur at 50°S and at 84 km altitude. Strong positive values are also observed above the equator between 65 and 70 km altitude, whereas the tropical region above this warming is characterized by an apparent cooling. An additional region with positive amplitudes occurs in the tropical lower stratosphere up to an altitude of 60 km in the southern hemisphere and between 60 – 75 km at 50°N. There is no evidence of a remaining signature potentially

related to the Pinatubo eruption with an amplitude of about 9 K, as previously reported (She et al., 1998).

Figure 4 shows the residuals of three more examples when no episodic term is fitted (equation 3, left column) and when this term is included (equation 2, right column). The time series at 84 km and 20 ± 5°N in the top row of Figure 4 exhibits high temperatures during the years 1991 to 1993 that are reduced, but still clearly visible, after fitting an episodic perturbation. Other locations in the upper mesosphere, as for example at 81 km altitude and 40 ± 5°N (middle row of Figure 4), do not exhibit

obviously enhanced temperatures in the first half of the 1990s for either fit. The example shown at the bottom of Figure 4 was chosen randomly. It shows the residuals at 79 km and 10 ± 5°N with slightly increasing temperatures at the beginning of the time series with the highest value in 1996. The first example in the top row is particularly interesting because the time series starts with high temperatures in the residual that start to fluctuate around 0 K after the beginning of 1993. More examples for residuals with unusually high temperatures at the beginning of the time series are presented in Figure S2 in the supplementary

material. Fitting the perturbation function proposed by a previous lidar study (She et al., 1998) does not specifically capture high temperatures at the beginning of the time series, i.e., in 1991/1992.

The left panel of Figure 5 corresponds to the top left panel in Figure 4 where the temperature residual without an episodic fit term is plotted for 84 km and 20°N. We propose a subsequent fit that describes the volcanic contribution as an exponential decay function with an e-folding time of 6 months (equation 4) because the HALOE instrument only started its scientific

measurements four months after the eruption and possibly missed parts of a potential mesospheric temperature signature. The fitted curve is shown as a blue line in the same graph together with the residuals. It captures the high temperatures of the data points at the beginning of the time series successfully. The temperatures at the beginning are reduced in the final residual (right side of Figure 5) after the fit is subtracted from the first residual (left side of Figure 5) which legitimates the use of this approach.

If the two fits are compared throughout the upper mesosphere, however, it is apparent that the inclusion of the episodic perturbation term adapted from the literature also captures the high temperatures at the beginning of the time series in some cases (column in the middle in Figure S1 of the supporting information). This term, however, strongly focuses on the time segment around 1993, whereas the exponential decay function specifically decreases the residual temperature at the start of

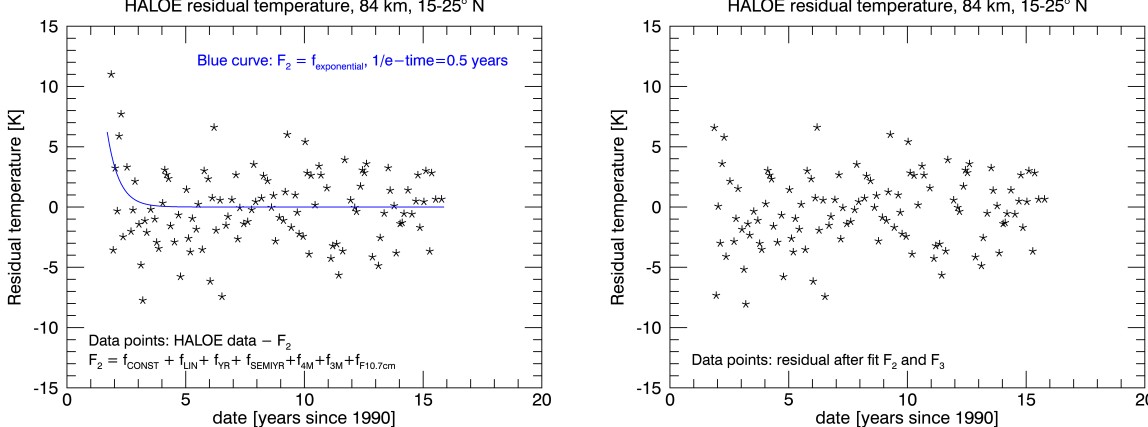

**Figure 5.** Implementing an exponential decay function as a volcanic fit term successfully captures data points with high temperatures at the end of 1991. The graph on the left side shows the temperature residual at 84 km and $20 \pm 5°$N after performing fit $F_2$ (that only accounts for a solar and several seasonal contributions but no volcanic term). High temperatures at the beginning of the time series clearly remain in the residual. An exponential decay function with an e-folding time of 0.5 years ($F_3$) was fitted to this residual and is shown as a blue curve. The graph on the right hand side presents the final residual after this curve was subsequently subtracted from the first residual and it demonstrates that the aforementioned high temperatures are reduced after this procedure.

the measurement series in most cases (last column in Figure S1 in the supporting information) and will therefore be explored
further.

To compare the alternative fit with the perturbation term adopted from the literature (She et al., 1998), Figure 6 shows the temperature of fit $F_3(t_{Oct91})$ in a contour plot with $t_{Oct91}$ being October 1991. We chose to report the value of the function at that time because it is more comparable with the amplitude of the perturbation term than the constant $C_0$ which is the intersection with the y axis, representing January 1990. The temperature of $F_3(t_{Oct91})$ is highest in the upper mesosphere above
170 75 km and between $10°$N and $30°$N with a maximum positive value of $5.4 \pm 3.0$ K, as can be seen in Figure 6. Another area with significant positive temperatures lies between 0 - $20°$S above 80 km. The lower mesosphere in the southern hemisphere shows positive values between $50 - 30°$S and $50 - 65$ km. As $t_{Oct91}$ is deliberately chosen as October 1991 it is, however, not quantitatively comparable to the amplitude of the perturbation function shown in the graph of Figure 3. Figure 6 nevertheless shows that, at least for the HALOE data set that started the observation four months after the Pinatubo eruption, an exponential
decay function is an alternative to the episodic perturbation term suggested by a previous study (She et al., 1998).

This conclusion is still valid when only the longitudes between $120°$W to $60°$W are considered that include the position of the lidar station at Fort Collins ($41°$N, $105°$W). The left side of Figure 7 shows the temperature anomalies according to an episodic perturbation similar to Figure 3, but for the restricted longitude range. Positive amplitudes above $40°$N are detected from $60 - 75$ km and even up to 80 km above $20 - 30°$N. The amplitude above 80 km in the northern hemisphere is still

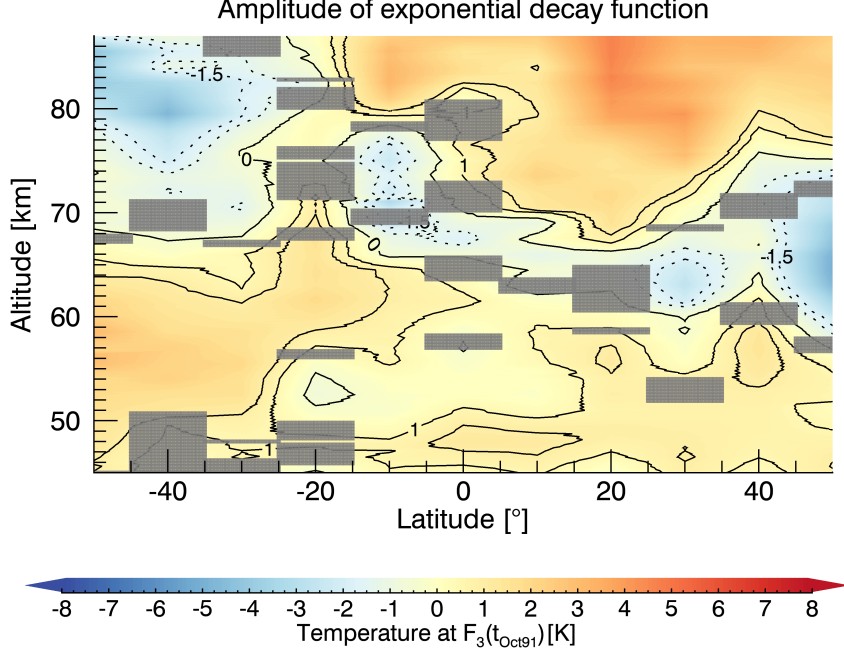

**Figure 6.** This contour plot shows the amplitude of the exponential decay function depending on the altitude and latitude. A positive temperature of $5.4 \pm 3.0$ K was found in the upper mesosphere above 80 km and at 20°N when an exponential decay function was applied. This temperature is the value of fit $F_3(t_{Oct91})$ at $t_{Oct91}$, i.e. October 1991, assuming that the peak of the perturbation occurred four months after the eruption.

negative. Using the exponential decay function (right side of Figure 7) suggests positive amplitudes in most of the upper mesosphere in the northern hemisphere with a maximum positive temperature at 40°N above 80 km of approximately 5 K.

The results thus indicate that a positive temperature anomaly is present in the tropical upper mesosphere at the beginning of the HALOE time series, which may be related to the eruption of Mt. Pinatubo. Possible mechanisms are discussed in the following section.

**4   Discussion**

This study compares two approaches to fit a perturbation signal potentially caused by the Pinatubo eruption in June 1991 to the HALOE temperature series in the middle atmosphere. They differ considerably in the assumed time that the volcanic perturbation needs to reach the upper mesosphere. Our HALOE study supported previous observations of an episodic warming in the upper mesosphere that might be related to the Pinatubo eruption in 1991. It indicates, however, that the mesosphere

was disturbed rapidly after the eruption and that the signature vanished quickly, whereas the Na lidar temperatures over Fort

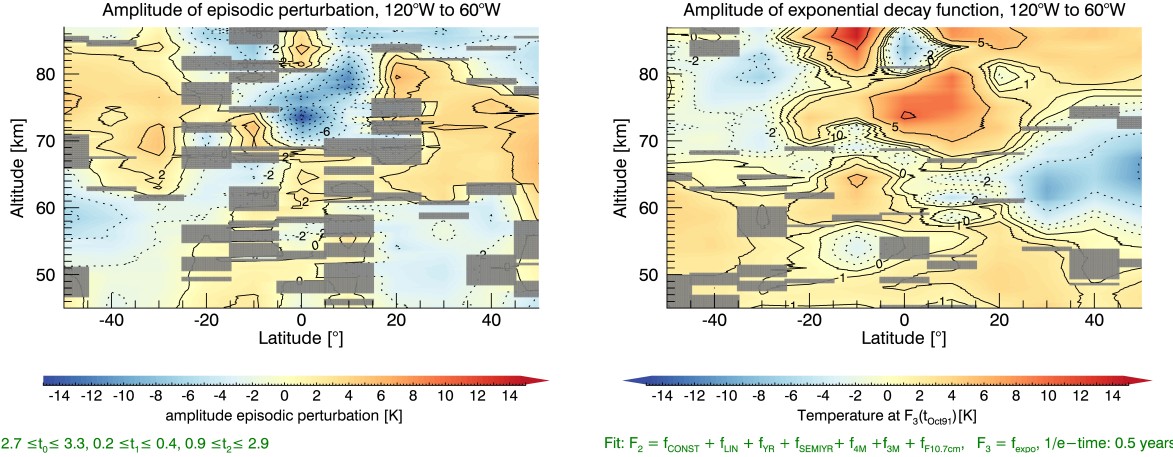

**Figure 7.** This figure shows temperature anomalies between $120°W$-$60°W$ for the case that an episodic perturbation described by She et al. (She et al., 1998) is used (left) and for the case that an exponentially decaying term is used to describe the volcanic signal (right hand side). This is the longitude bin that includes the position of the lidar station at Fort Collins ($41°N$, $105°W$). Using the exponentially decaying function, the maximum temperature at $40°N$ and above 80 km is $\approx 5$ K. This temperature is the value of fit $F_3(t_{Oct91})$ at $t_{Oct91}$, i.e. October 1991, assuming that the peak of the perturbation occurred four month after the eruption.

Collins ($40.6°N$, $105.1°W$) showed a strong mesospheric warming peaking in 1993 (She et al., 1998). There are, however, uncertainties in comparing a ground-based measurement with zonally averaged satellite data using a $10°$ latitude bin for the case of the HALOE analysis. Nevertheless, both instruments have a similar vertical resolution of about 3 km (She et al., 1998) and 3.5 km (Remsberg, 2009) for the lidar and HALOE profiles, respectively. Keckhut et al. (1995) reported Rayleigh lidar measurements at $44°N$ from $30-80$ km altitude and observed a mesospheric warming in the summer of 1992 and 1993 below 80 km. At their measurement top height in 80 km altitude, however, their analysis suggests a cooling for both summers which is in disagreement with the observation of She et al. (1998). Some observational studies covering the mesosphere, detect a temperature perturbation within half a year after the eruption and see a rapid decline of this signal. Analysis of OH*(3-1) rotational temperatures acquired in Wuppertal and Northern Scandinavia (Andoya at $69°$ N and Kiruna at $68°$ N) revealed a potential Pinatubo related signal in the phases of the fitted seasonal parameters (Bittner et al., 2002). They appear 6 months after the eruption and vanished half a year later (Bittner et al., 2002). Another publication used the OH*(3-1) rotational temperatures from both Wuppertal and Hohenpeißenberg and showed a positive anomaly in 1991 in fitted mean temperatures that disappeared in 1992 (Offermann et al., 2010). It has to be pointed out that the results published in the literature vary in the spatial range that is discussed (zonally averaged results compared to local ground-based measurements) and that some only focus on a specific season while others do not make such a separation. Bittner et al. (2002) speculated that the Pinatubo-induced temperature signal in the mesosphere might not be homogeneous in the zonal direction so that ground-based measurements differ because of a zonal asymmetry. The data sampling of HALOE is spatially sparse because of the limitation of the solar

occultation geometry, therefore zonal averages are preferably discussed in the literature. If only longitudes between 60 – 120°W are averaged, a similar difference between the two approaches is apparent for the upper mesosphere above 80 km in the northern hemisphere. This implies that the analysis of the HALOE data presented here does not support the hypothesis that the differences between our results and the results of She et al. (1998) can be explained by a zonal asymmetry in the response of the middle atmosphere to the Pinatubo eruption.

It is nontrivial to separate a Pinatubo-related signature from other natural contributions (such as a solar signal (Kerzenmacher et al., 2006)) when analysing temperature data and this issue could potentially explain some of the disagreements in the observations. The 11-year solar cycle exhibited a maximum around 1990 with several solar flares occurring in June 1991 (Rank et al., 1994). Remsberg (2009) demonstrated that an 11-year oscillation fitted to the HALOE temperature data between 40°S - 40°N in the middle atmosphere is approximately in phase with solar UV flux measurements for most altitudes. We therefore use the F10.7 cm solar flux as a solar proxy in our fit routine and limit its amplitude to positive values including zero. This should capture the contribution from the 11-year solar cycle in order to separate its effect from the volcanic terms. Another type of interfering signal was observed for OH* temperatures in the mesopause when Kalicinsky et al. (2016) found a 25-year oscillation of unknown origin with an amplitude of (1.95±0.44) K that has a maximum in 1993. They saw both the 25 year oscillation and a temperature peak in 1991 that coincided with the Pinatubo eruption. These findings emphasize that any solar or non-solar signal must be carefully separated from a Pinatubo perturbation for each temperature time series.

First results of a tropical volcanic eruption, that injects twice as much $SO_2$ as the Pinatubo, with the upper-atmosphere icosahedral non-hydrostatic (UA-ICON) model (Borchert et al., 2019) suggest that the strongest response in the upper mesosphere appears approximately 6 months after the eruption (Hauke Schmidt, personal communication, August, 24 2021). Interestingly, it also started to fade away and is barely visible two years after the eruption which further supports the hypothesis that volcanic perturbations can rapidly reach the mesosphere. Although the simulation showed the strongest perturbation in December we decided to report the value of $F_3(t)$ for October because single time series show high residual temperatures before the month of December. Both observation and simulation provide evidence that support our assumption of an early temperature signal in the mesosphere. As the HALOE instrument only started operating four months after the eruption, it is suggested that it may only detect the decay of the already fading signal.

An increase in the mesospheric temperature due to a strong volcanic eruption would indicate a coupling mechanism that transports the signal in the lower stratosphere up to the mesosphere. First simulations of the aforementioned UA-ICON model hint at a dynamic mechanism where the anomalous warming of the tropical lower stratosphere by the volcanic aerosols increases the meridional temperature gradient. This would affect the zonal wind because of the thermal wind relation and could induce changes in the filtering of gravity waves. As a consequence, the wave-driven residual circulation in the upper mesosphere is altered leading to adiabatic cooling/heating and thus to a temperature anomaly. An in-depth study of this mechanism by the UA-ICON model is currently on the way. Previous simulations are in agreement with the overall argument of our proposed mechanism. A simulation performed by Rind et al. (1992) showed that the radiative heating of the tropical lower stratosphere after a volcanic eruption enhanced the static stability of the troposphere in the low and mid-latitudes. These atmospheric conditions enabled an increased wave propagation to the stratosphere that subsequently strengthened the lower stratospheric branch

of the Brewer-Dobson circulation. The connected enhancement of the upward velocity induced a cooling in the tropics from the middle stratosphere to the mesosphere and the downward movement at the poles likewise caused a warming of the up-per stratosphere and lower mesosphere. Importantly, the model also exhibited increased temperatures in the upper part of the mesosphere. The propagation of thermal and/or dynamical perturbations to the mesosphere and mesopause region is mediated by gravity waves. This coupling is relatively fast and occurs on time-scales of days. Becker and von Savigny (2010) reported on the graviy wave-driven thermal effects at the polar summer mesopause caused by thermal perturbations of the lower polar mesosphere that stem from solar proton events. The thermal response at the polar summer mesopause occurred within a few days. Karlsson et al. (2009) demonstrated that the perturbation associated with a sudden stratospheric warming in the polar winter stratosphere only takes a few days to reach the polar summer mesopause region. In addition, a rapid dynamic coupling was reported, e.g. by Smith and Mullen (2020) who used WACCM6 simulation to demonstrate that perturbations in the winter stratosphere impact the summer mesosphere via a wave-driven inter-hemispheric coupling in just a few days.

## 5    Conclusions

The HALOE temperature time series for the mesosphere region was analyzed for a potential perturbation caused by the eruption of Mount Pinatubo in June 1991. The data was fitted with a regression that accounted for a constant and linear term, annual, semi-annual, 4-month and 3-month oscillations, as well as the F10.7cm solar proxy. Two methods to estimate a potential volcanic signal were compared. The first expanded the regression and included a perturbation term first proposed in a lidar study (She et al., 1998) that reported an episodic warming occurring in 1993. Using the parameters reported in that lidar study for 86 km and comparing it to HALOE data at a similar altitude and latitude resulted in an amplitudes with opposite sign; indicating a cooling instead of a warming. HALOE data for the entire mesosphere was subsequently fitted using ranges for the perturbation parameters $t_0$ to $t_2$ that were defined by the values found and discussed in the aforementioned publication (She et al., 1998). A maximum positive amplitude of $4.1 \pm 1.4$ K was observed and supported the finding of the previous study that a temporary warming seemed to occur in the upper mesosphere region after the Pinatubo eruption. Our estimated temperature signal, however, seems to be lower than the one reported by She et al. (1998). Differences are nevertheless expected because a ground-based measurement is compared with zonally averaged occultation measurements using a $10°$ latitude bin. Applying the aforementioned fit to the HALOE data between 60-120° W (including the longitude of the Fort Collins lidar station) does not substantially differ from the results for the global zonal mean.

Our analysis revealed anomalous positive temperature anomalies at some latitudes in the upper mesosphere during the first months of the HALOE time series starting in October 1991. For this reason, a second fit method was applied to fit the de-seasonalized data with an exponential decay function having an e-folding time of 6 months. This approach suggests a volcanic warming of up to $5.4 \pm 3.0$ K if the peak of the signature is assumed to be reached in October 1991, four months after the eruption. It indicates that HALOE probably measures (only) the decay of a mesospheric perturbation that was forced by the Pinatubo eruption and also suggests a rapid response of the mesosphere to the volcanic event in agreement with other observations (Kalicinsky et al., 2016; Offermann et al., 2010).

*Data availability.* The HALOE level 2 version 19 temperature data used in this paper is available on the NASA website (https://disc.gsfc.nasa.gov, Russell III (1999)). Three IDL routines were used for the subsequent data analysis, namely the IDL routine MPFUNFIT.PRO that was released by Craig Markwardt (http://cow.physics.wisc.edu/craigm/idl/fitting.html), PERCENTILES.PRO by Martin Schultz (https://hesperia.gsfc.nasa.gov/ssw/packages/s3drs/idl/util/percentiles.pro) and the IDL reading function READ_HALOE_L2.PRO that was obtained from NASA (https://mls.jpl.nasa.gov/data/readers.php). The Laboratory for Atmospheric and Space Physics (LASP) provided the F10.7cm solar flux that is used as a solar proxy in this paper (http://lasp.colorado.edu/lisird/data/penticton_radio_flux/).

## Appendix A

### A1    Calculating confidence intervals and setting a significance criterion

The 95% confidence interval of the regression coefficients was determined by a delete-1 jackknife method (Miller, 1974). For each time series at a specific altitude and latitude bin of the size n, an (n-1)-sized sub-sample was drawn n times, creating n sub-samples were each measurement value is omitted once. This imitates n hypothetical measurement of the time series were one measurement value is missing in every observation compared to the original one. The regression coefficients were determined for every sub-sample. All results were sorted from lowest to highest and the values marking 2.5% and 97.5% of the distribution, respectively, were taken as the boundary of the 95% confidence interval. The regression coefficient is regarded as significant if the 95% confidence interval of the distribution from the sub-sampling did not include zero.

*Author contributions.* CvS outlined the project. SW carried out the analysis and wrote an initial version of the manuscript. All authors discussed, edited and proofread the paper.

*Competing interests.* The authors declare that they have no competing interests.

*Acknowledgements.* This study is part of the VolDyn project that is embedded in the research unit VolImpact (grant no. 398006378) and funded by the German Research Foundation (DFG).

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
