# Peer review of "Estimating the impact of the 1991 Pinatubo eruption on mesospheric temperature by analyzing HALOE (UARS) temperature data"

_Annales Geophysicae, 2021_

## Referee Comment (RC1)

Comments on ANGEO manuscript 2021-50

                  By Ellis Remsberg

I have several concerns and suggestions for the manuscript version under review, and I have already passed them along to the authors privately.  Their reply to me indicates that accounting for anomalous

HALOE profiles makes a quantitative difference for their findings, but does not change their conclusion that there may be some effect on temperature in late 1991 from the Pinatubo eruption.  However, I do not think that the Pinatubo effects extend to 1993.  The following paragraphs give my comments on the version of the manuscript that is currently under review.  I am sure that the authors will make appropriate revisions based on their updated findings.

My past work on analyzing HALOE temperatures was exploratory, initially, because I was looking also for the Pinatubo effect that was alluded to by She et al. (1998).  I was concerned about decadal-scale, dynamical influences, so I decided not to regress against a solar flux proxy, but to fit temperature series with an 11-yr sinusoid and check about its phasing with that of a solar cycle proxy.  Subsequent to

Remsberg and Deaver (2005), I discovered that I had not accounted properly for autoregressive effects.

I reported on that realization and then updated my analyses in Remsberg (2007).  I also used that revised approach in the analyses of Remsberg (2008; 2009).  Most recently, I reported on analyses of

HALOE water vapor in the mesosphere, and I included further updates about HALOE temperature, as well (Remsberg et al., 2018).  For that study, I included a regression term for a solar proxy (Lyman-alpha flux) and a dynamical term related to an ENSO proxy (MEI index), and I obtained a good fit to the H2O

time series and an improved fit to the temperature time series.  However, I did not include a volcanic aerosol proxy term because I did not see that one was indicated from the time series of the model/data residuals.

After reading your manuscript, I re-analyzed HALOE temperature time series at 0.03 hPa in the manner of Remsberg et al. (2018), see Figure 1 below.  In that paper I noted that there are so-called 'trip angle'

biases in many of the SR profiles of November 1991 and April 1992; that problem is described in more detail on the GATS, Inc., HALOE Website homepage, including which profiles are not trustworthy.  The regression analysis for Fig. 1 begins in October 1991 and includes SR and SS profiles at 37N +/- 7.5

degrees of latitude, which overlaps the sodium lidar station at Fort Collins (41N); you can compare them with that of Remsberg and Deaver (2005, Fig. 6).  The response in Fig. 1 of HALOE T(p) to the Lyman- alpha flux proxy (max-min) is +1.8 K, and the analyzed linear trend for T(p) is -2.4 K/decade.

Figure 2 is my analysis for T(p) at 0.01 hPa or nearer to the sodium layer viewed by She et al.  The response of T(p) to Lyman-alpha flux is +3.3 K, but the linear trend is now weakly positive (+0.46

K/decade) although it is not significant.  In addition, my analyzed coefficient for the ENSO index is 0.86

K/MEI index, and it is highly significant.  This pressure altitude of 0.01 hPa is where the dissipation of gravity waves (and the phase of ENSO) may be affecting temperature.   Again, my regression model fits
the data well, even during the year or so following the Pinatubo eruption.

Finally, Figure 3 is an "image" of the longitude/pressure cross section for SS profiles on 15 December
1991 at 38N latitude.  Note that HALOE viewed this latitude only about once every month or so.  You
may access and view similar images at the HALOE Website by clicking on 'Browse images' on the left
menu of the homepage, and then selecting from the next pop-up page, a longitude/pressure plot, the
parameter of interest (temperature), the months of a given year, and whether you wish to look at SS
and/or SR scans.  When you make your request at the bottom of that Webpage, the list of days that you
asked for appears along with their mean latitude; clicking on one particular day then allows you to view
more images.  I show Fig. 3, so that you will see that there was a region of warm temperatures (225 K)
near 0.01 hPa and about 270 E longitude (~Ft. Collins).  There is also a pronounced zonal wave-1 in T(p)
at this pressure level (see the cold values of ~200K near 60E).  To my mind, this longitudinal variation in
T(p) indicates effects from the interaction of planetary waves with gravity waves in the upper
mesosphere.  It may be that there is an indirect connection with the Pinatubo event, too, wherein its
aerosol layers have altered slightly the meridional T(p) gradient of the lower stratosphere and allowed
for a redirection of the transmission of gravity waves to the mesosphere.  On the other hand, transport
times and radiative relaxation rates are short in the upper mesosphere, such that any temperature
anomalies decay quickly.  A sustained transmission of gravity waves would be necessary to maintain
those anomalies, in my opinion.

Additional references:

Remsberg, E. E., A re-analysis for the seasonal and longer-period cycles and the trends in middle
atmosphere temperature from HALOE, J. Geophys. Res.—Atmospheres, vol. 112, D09118,
doi:10.1029/2006JD007489, 2007.

Remsberg, E. E., On the response of Halogen Occultation Experiment (HALOE) stratospheric ozone and
temperature to the 11-yr solar cycle forcing, J. Geophys. Res.-Atmospheres, 113, D22304,
doi:10.1029/2008JD010189, 2008.

Remsberg, E., Damadeo, R., Natarajan, M., and Bhatt, P.: Observed responses of mesospheric water
vapor to solar cycle and dynamical forcings, J. Geophys. Res., 123, 3830-3843,
https://doi.org/10.1002/2017JD028029, 2018.

[Figure]

Figure 1—HALOE temperature time series at 0.03 hPa and 37N +/- 7°.  Regression model terms are listed
at lower left; x-axis is days since 1 January 1991.

[Figure]

Figure 2—As in Fig. 1, but for 0.01 hPa.

[Figure]

Figure 3—Image of longitude-pressure cross section for HALOE T(p) at SS for 15 December 1991.

---

## Author Response (AR1)

Dear Dr. Gunter Stober,

thank you very much for giving us the opportunity to submit a revised draft of the manuscript "Estimating the impact of the 1991 Pinatubo eruption on mesospheric temperature by analyzing HALOE (UARS) temperature data" to Annales Geophysicae. We appreciate the time and effort that you and the reviewers dedicated to providing feedback on our manuscript and are grateful for the insightful comments on our paper that resulted in its improvement. We have highlighted the changes within the manuscript.

There are some minor changes in the revised manuscript that where not directly requested by the reviewers, but that we made in order to clarify some of the statements.
In line 34 we added: "…compared to the observed time period of 1992 – 1998 ...".
In line 45 we rephrased one of our sentences to "they did not consider a volcanic contribution for their analysis and only reported a maximum of an 11 year solar cycle-like term around 1993".
We clarified some details of the data analysis further in line 81 ("if at least two measurements existed for this time/latitude bin"), line 87 ("if at least two measurements existed for this bin and month") and in line 104 ("the regression coefficient $B_{10}$ of the solar proxy term"). We also added minor changes to the result section for clarity, i.e. in line 115 ("This study uses a data product where the HALOE measurements from 76.5 km to 99 km have been merged with the MSIS (Mass Spectrometer – Incoherent Scatter) climatology during the processing."), line 124 ("(maximum or minimum value)") and line 129 ("In contrast to $A_{11}$").
One paragraph was shifted downwards to line 222 to strengthen the argument.
An additional sentence was added to Figure 6 for greater readability. We also included a reference to the PERCENTILES.PRO routine to the data availability section in line 276.
Finally, we shortened one phrase in Figure 1 of the supplemental material from 'Its amplitude is often not significant or suggests a post-volcanic cooling of the upper mesosphere' to 'Its amplitude often suggests a post-volcanic cooling of the upper mesosphere' due to a revised analysis (please see below).

We also revised our analysis after Dr. Ellis Remsberg pointed out that some HALOE measurements are not advised for scientific use because of problematic lockdown or trip angles. All figures and associated amplitudes that are reported in this manuscript are revised accordingly.
Finally, after replying to the comments of Dr. Philippe Keckhut, we changed the method for determining the uncertainty and report here the final approach that we use in the resubmitted manuscript.

**Answers to the review from Dr. Ellis Remsberg**

Comments on ANGEO manuscript 2021-501
By Ellis Remsberg

We would like to thank the reviewer for taking the time to assess our manuscript. We will address all of the reviewer's comments in the following paragraphs.

I have several concerns and suggestions for the manuscript version under review, and I have already passed them along to the authors privately. Their reply to me indicates that accounting for anomalous HALOE profiles makes a quantitative difference for their findings, but does not change their conclusion that there may be some effect on temperature in late 1991 from the Pinatubo eruption. However, I do not think that the Pinatubo effects extend to 1993. The following paragraphs give my comments on the version of the manuscript that is currently under review. I am sure that the authors will make appropriate revisions based on their updated findings.

We would like to thank the reviewer for drawing our attention to the HALOE measurements that are tagged as unreliable because of problematic trip and/or lockdown angles. We revised our analysis and inserted a short sentence in the 'data and data  analysis' section of the manuscript from line 78-79: "Measurements that were tagged as being problematic due their corresponding trip or lockdown angle (http://haloe.gats-inc.com/user_docs/index.php) were dismissed from the analysis."

My past work on analyzing HALOE temperatures was exploratory, initially, because I was looking also for the Pinatubo effect that was alluded to by She et al. (1998). I was concerned about decadal-scale, dynamical influences, so I decided not to regress against a solar flux proxy, but to fit temperature series with an 11-yr sinusoid and check about its phasing with that of a solar cycle proxy. Subsequent to Remsberg and Deaver (2005), I discovered that I had not accounted properly for autoregressive effects. I reported on that realization and then updated my analyses in Remsberg (2007). I also used that revised approach in the analyses of Remsberg (2008; 2009). Most recently, I reported on analyses of HALOE water vapor in the mesosphere, and I included further updates about HALOE temperature, as well (Remsberg et al., 2018). For that study, I included a regression term for a solar proxy (Lyman-alpha flux) and a dynamical term related to an ENSO proxy (MEI index), and I obtained a good fit to the $H2O$ time series and an improved fit to the temperature time series. However, I did not include a volcanic aerosol proxy term because I did not see that one was indicated from the time series of the model/data residuals.

We thank the reviewer for his comment on the necessity of a volcanic proxy to fit the HALOE temperature data. Our analysis hints to unusually high temperatures at the beginning of the time series only in some altitude/latitude segments of the mesosphere. These are the focus of this manuscript and seem to suggest a warming that might be related to the prior eruption of Mt. Pinatubo. We are nevertheless aware that no direct correlation between the Pinatubo eruption and a warming of the mesosphere was proven by our study, but that it might add a small contribution to the ongoing discussion in the community.

After reading your manuscript, I re-analyzed HALOE temperature time series at 0.03 hPa in the manner of Remsberg et al. (2018), see Figure 1 below. In that paper I noted that there are so-called 'trip angle' biases in many of the SR profiles of November 1991 and April 1992; that problem is described in more detail on the GATS, Inc., HALOE Website homepage, including which profiles are not trustworthy. The regression analysis for Fig. 1 begins in October 1991 and includes SR and SS profiles at 37N +/- 7.5 degrees of latitude, which overlaps the sodium lidar station at Fort Collins (41N); you can compare them with that of Remsberg and Deaver (2005, Fig. 6). The response in Fig. 1 of HALOE T(p) to the Lyman- alpha flux proxy (max-min) is +1.8 K, and the analyzed linear trend for T(p) is -2.4 K/decade.

We again thank the reviewer for bringing our attention to the existence of the trip and lockdown angle problem. We revised the data analysis (see answer given above).

Figure 2 is my analysis for T(p) at 0.01 hPa or nearer to the sodium layer viewed by She et al. The response of T(p) to Lyman-alpha flux is +3.3 K, but the linear trend is now weakly positive (+0.46 K/decade) although it is not significant. In addition, my analyzed coefficient for the ENSO index is 0.86 K/MEI index, and it is highly significant. This pressure altitude of 0.01 hPa is where the dissipation of gravity waves (and the phase of ENSO) may be affecting temperature. Again, my regression model fits the data well, even during the year or so following the Pinatubo eruption.

Finally, Figure 3 is an "image" of the longitude/pressure cross section for SS profiles on 15 December 1991 at 38N latitude. Note that HALOE viewed this latitude only about once every month or so. You may access and view similar images at the HALOE Website by clicking on

'Browse images' on the left menu of the homepage, and then selecting from the next pop-up page, a longitude/pressure plot, the parameter of interest (temperature), the months of a given year, and whether you wish to look at SS and/or SR scans. When you make your request at the bottom of that Webpage, the list of days that you asked for appears along with their mean latitude; clicking on one particular day then allows you to view more images. I show Fig. 3, so that you will see that there was a region of warm temperatures (225 K) near 0.01 hPa and about 270 E longitude (~Ft. Collins). There is also a pronounced zonal wave-1 in T(p) at this pressure level (see the cold values of ~200K near 60E). To my mind, this longitudinal variation in T(p) indicates effects from the interaction of planetary waves with gravity waves in the upper mesosphere. It may be that there is an indirect connection with the Pinatubo event, too, wherein its aerosol layers have altered slightly the meridional T(p) gradient of the lower stratosphere and allowed for a redirection of the transmission of gravity waves to the mesosphere. On the other hand, transport times and radiative relaxation rates are short in the upper mesosphere, such that any temperature anomalies decay quickly. A sustained transmission of gravity waves would be necessary to maintain those anomalies, in my opinion.

We would like to thank the reviewer for taking the time to provide the plots that he shared. Our study so far considered zonal mean temperatures. We included a new figure (Figure 7 in the revised version) where we present the results of our two fitting approaches for measurement data sampled between 60°-120°W. This region includes the location of the lidar station in Fort Collins. The plots with the limited longitude range supports the overall argument of our paper and we added it together with a short description to the manuscript, beginning from line 174: "This conclusion is still valid when only the longitudes between 120°W to 60 °W are considered that include the position of the lidar station at Fort Collins (41°N, 105°W). The left side of Figure 7 shows the temperature anomalies according to an episodic perturbation similar to Figure 3, but for the restricted longitude range. Positive amplitudes above 40°N are detected from 60 – 75 km and even up to 80 km above 20 – 30°N. The amplitude above 80 km in the northern hemisphere is still negative. Using the exponential decay function (right side of Figure 7) suggests positive amplitudes in most of the upper mesosphere in the northern hemisphere with a maximum positive temperature at 40°N above 80 km of approximately 5 K."

Additional references:
Remsberg, E. E., A re-analysis for the seasonal and longer-period cycles and the trends in middle atmosphere temperature from HALOE, J. Geophys. Res.—Atmospheres, vol. 112, D09118, doi:10.1029/2006JD007489, 2007.

Remsberg, E. E., On the response of Halogen Occultation Experiment (HALOE) stratospheric ozone and temperature to the 11-yr solar cycle forcing, J. Geophys. Res.-Atmospheres, 113, D22304, doi:10.1029/2008JD010189, 2008.

Remsberg, E., Damadeo, R., Natarajan, M., and Bhatt, P.: Observed responses of mesospheric water vapor to solar cycle and dynamical forcings, J. Geophys. Res., 123, 3830-3843, https://doi.org/10.1002/2017JD028029, 2018

We would like to thank the referee again for taking the time to review our manuscript and for providing additional graphics that added value to the ongoing discussion of volcanic-driven signals in the middle atmosphere in the community.

**Answers to the review from Dr. Philippe Keckhut**

We would like to thank the reviewer for taking the time to assess our manuscript. In the following paragraphs, we will address all of the reviewer's comments.

This study concerns the quantification of the mesospheric impact of the Pinatubo eruption using the HALOE instrument on board UARS platform.

While these series miss the beginning of the event, it is interesting to perform such analyses because Pinatubo is one of the biggest eruptions observed in the last decades that has perturbed the whole atmosphere.

This analysis confirms previous analyses that have indicated a warming of the mesosphere following the eruption and deserve to be published. However, I think the amplitude estimates provide in this study needs to be carefully discussed while this study do not provide any uncertainties.

Thank you very much for this comment. To address this, uncertainties for the amplitudes were determined as the standard error from the jackknife estimate. The uncertainties were added to the reported amplitudes and a short sentence explaining the procedure was included from line 107-108: "The uncertainties for the amplitudes are reported as the standard errors of the jackknife estimate (Efron, 1981)."

Efron, Bradley. "Nonparametric Estimates of Standard Error: The Jackknife, the Bootstrap and Other Methods." Biometrika, vol. 68, no. 3, 1981, pp. 589–99, https://doi.org/10.2307/2335441

The conclusion that there is a discrepancy with previous studies seems then to be too strong. The conclusion should be more positive while warming was confirmed.

We agree with the reviewers opinion, that the manuscript should also highlight the aspects that are in agreement with the study of She et al. (1998) that we cite. We added to our discussion, line 186-187: "Our HALOE study supported previous observations of an episodic warming in the upper mesosphere that might be related to the Pinatubo eruption in 1991. " and included additional sentences to our conclusion from line 261-263: "A maximum positive amplitude of 4.1 ± 1.4 K was observed and supported the finding of the previous study that a temporary warming seemed to occur in the upper mesosphere region after the Pinatubo eruption. Our estimated temperature signal, however, seems to be lower than the one reported by She et al. (1998)."

The volcanic eruptions are complex to quantify while solar cycles match the occurrence of volcanic eruptions mainly when series are short (see for example Kerzenmacher et al., 2006).

We thank the reviewer for drawing our attention to the publication of Kerzenmacher et al. (2006). We added this citation to our discussion about the problem of separating the volcanic from the solar contributions to the mesospheric temperature in line 211.

Temperature deviations associated with Pinatubo eruptions are calculated with zonal average while other estimates are local observations. Also some estimates are performed by season and some other including all season. If a dynamical effect is expected, Pinatubo signature should be different.

We agree with the reviewer and added the following sentence to the discussion from line 201 -203: "It has to be pointed out that the results published in the literature vary in the spatial range that is

discussed (zonally averaged results compared to local ground-based measurements) and that some only focus on a specific season while others do not make such a separation."

Also data quality needs to be discussed either the absolute values (see Remsberg et al., 2002) and the number of data while solar occultations provide a smaller sampling than more traditional observation like nadir observations while the vertical resolution is better.

In order to help with the comparison of the results from the satellite-borne HALOE and the ground-based lidar instrument, we added a comment on their similar vertical resolution in line 191: "Nevertheless, both instruments have a similar vertical resolution of about 3 km (She et al., 1998) and 3.5 km (Remsberg, 2009) for the lidar and HALOE profiles, respectively."

Remsberg, E. E.: Trends and solar cycle effects in temperature versus altitude from the Halogen Occultation Experiment for the mesosphere and upper stratosphere, Journal of Geophysical Research: Atmospheres, 114, https://doi.org/10.1029/2009JD011897, 2009

Another global estimate with a different dataset can be used for comparison and can be found in Hampson et al. (2006).

We thank the reviewer for pointing out the TOVS data set used in Hampson et al. (2006). TOVS temperatures are available up to 10 mbar altitude, i.e. up to the middle stratosphere (Scott et al., 1999). Although another data set based on TOVS exists that even provides temperature layers between 1 to 0.4 mbar (https://daac.ornl.gov/FIFE/Datasets/Atmosphere/TOVS_atmos_prof.html), this as well only covers the upper stratosphere and lower mesosphere. Since our study focuses on the impact of the Pinatubo eruption on the mesosphere and mesopause region, this would be out of the scope of our study.

Scott, N. A., Chédin, A., Armante, R., Francis, J., Stubenrauch, C., Chaboureau, J., Chevallier, F., Claud, C., & Cheruy, F. (1999). Characteristics of the TOVS Pathfinder Path-B Dataset, Bulletin of the American Meteorological Society, *80*(12), 2679-2702

Kerzenmacher et al., Methodological uncertainties in multi-regression analyses of middle-atmospheric data series, 2006, *J. Environ. Monit.*, 8, 682-690, DOI:10.1039/b603750j.

Hampson et al., The dynamical influence of the Pinatubo eruption in the subtropical stratosphere, *J. Atmos. Solar Terr. Phys.*, 68, 1600-1608, 2006, dx.doi.org/10.1016/j.jastp.2006.05.009.

Remsberg et al., An assessment of the quality of HALOE temperature profiles in the mesosphere with Rayleigh backscatter lidar and inflatable falling sphere measurements, *J. Geophys. Res.*, 107(D19), 10.129/2001jD001521, 2002.

We thank the reviewer again for taking the time to revise our manuscript and for providing insightful comments.

**Answers to the review of the anonymous referee**

We would like to thank the reviewer for taking the time to assess our manuscript.

In this manuscript, the authors studied the impact of Pinatubo volcanic eruption on the mesospheric temperature obtained from the HALOE instrument. They obsered warming in the mesosphere and compared with the Na lidar temperature results published earlier. As the number of profiles is less in HALOE observations, it is difficult to believe the results.

We appreciate the reviewer's comment and would like to elaborate on the purpose of this study. The Pinatubo eruption was the largest volcanic eruption in the past 40+years (in terms of $SO_2$ mass injected into the stratosphere and also regarding its effects on surface and lower stratospheric temperature perturbations). There are very few data sets of middle atmospheric temperature covering the period around or after the eruption. The HALOE data sets is one of these data sets and it has not yet been specifically analyzed to search for potential temperature perturbations associated with the Mt. Pinatubo eruption. For these reasons we consider it worthwhile to do this.
We don't fully understand the reviewer's comment that the number of profiles is less in HALOE observations. The Na-lidar temperature observations analyzed by She et al., for example, relied on 4 and 5 nights of measurements during the springs of 1990 and 1991. After May 1991 the number of measurements increased to 4 to 5 nights a month. We would also like to point out that the OH temperature observations carried out at Wuppertal also show an anomalous positive temperature enhancement in the months after the Pinatubo eruption. This strengthens the need for further studies on the impact of the Pinatubo eruption on the mesosphere, such as is provided by our study.

As the eruption occurred much earlier before the HALOE observations start, the state of the mesosphere prior to the eruption and the evolution and decay of the temperature perturbations could not be captured. The authors may use TIMED SABER or similar temperature data sets for any recent major volcanic eruptions to further strengthen the result.

We agree that an investigation of recent major eruptions with the SABER data would be generally insightful, but (almost) all of the eruptions that occurred during the TIMED-SABER time period were associated with $SO_2$ injections into the stratosphere that were at least a factor of 10 smaller than for the Pinatubo eruption. None of these eruptions produced such an obvious response in surface temperature or lower stratospheric temperature and we believe that the identification of a potential volcanic signature in middle atmospheric temperatures will be very challenging for these eruptions if not impossible. We have chosen Pinatubo, because it led to a relatively strong thermal perturbation of the lower tropical stratosphere.

Also, the authors need to explan how the volcanic eruption can cause warming in the mesosphere.

Thank you for this comment. We added a description of a dynamical mechanism that can explain the mesopause warming as a consequence of the lower stratospheric and low latitude diabatic heating by the Pinatubo aerosol layer. Please note that we have first simulation results with the upper atmosphere version of the ICON model that are consistent with this mechanism. It involves an anomalous warming of the tropical lower stratosphere (by the volcanic aerosols), hence an increase in the meridional temperature gradient; this in turn affects the zonal wind because of the thermal wind relation; a change in zonal winds leads to a change in the filtering of gravity waves and in consequence affects the wave-driven residual circulation in the upper mesosphere. At last, a temperature anomaly is caused in the mesopause region by a change in adiabatic cooling/heating. We added a description of the mechanism to the discussion section.

We would like to thank the referee again for taking the time to review our manuscript.

---

## Author Response (AR2)

Dear Dr. Gunter Stober,

thank you very much for giving us the opportunity to submit a revised draft of the manuscript "Estimating the impact of the 1991 Pinatubo eruption on mesospheric temperature by analyzing HALOE (UARS) temperature data" to Annales Geophysicae. We appreciate the time and effort that you and the reviewers dedicated to providing feedback on our manuscript and are grateful for the insightful comments on our paper that resulted in its improvement. We have highlighted the changes within the manuscript.

**Answers to the review from Dr. Ellis Remsberg**

You have conducted an extensive set of analyses that include an episodic term to simulate the possible effects of the Pinatubo eruption on temperature trends in the upper mesosphere. Below I have several comments and/or suggestions that would improve your manuscript.

We would like to thank the reviewer for taking the time to assess our manuscript and agree that his comments resulted in its improvement. We will address all of the reviewer's comments in the following paragraphs.

Lines 42 to 46—It is not quite true that "Remsberg (2009) did not consider…". I suggest merely saying "did not include,,,". In my earlier review I pointed to my most recent model for fitting the HALOE temperature time series (Remsberg et al., 2018), and I achieved good fits by including a solar flux proxy term and, more importantly, a term related to the ENSO forcing (MEI). You may find that a model that includes an MEI term will yield different results.

We thank the reviewer for clarifying this point and changed the manuscript as he suggested.

Line 119—Your use of the episodic term would be more convincing if you first showed the fit and residual at 35-45°N and 86 km using your model without the episodic term. Perhaps, you could at least put it in the supplement. Then, after the reader is aware of your result without the episodic term, you could then show your Figures 1 and 2 for comparison.

We agree with the reviewer and added two graphs to the supplementary material (now indicated as Figure S1) showing the fit and the residuals without a volcanic term. The following sentence is added to line 110-113: "We fit the temperature data and compare two approaches that each include a term for a potential volcanic perturbation (for the results of the fit without any volcanic contribution, please see Figure S1 in the supplementary material and compare them with Figure 1 and Figure 2)."

Line 125—Is A11 the amplitude?

$A_{11}$ is one of the regression coefficients in fit F1(t) and is part of the term that describes the volcanic perturbation. As the perturbation function described by this term is not normalized to one, we do not report the value of $A_{11}$ as the amplitude of the perturbation signal. Rather the maximum or minimum value of the function that is described by this term is used as the amplitude. We added a short phrase to the sentence in line 131 for a better understanding: "In contrast to the regression coefficient $A_{11}$, …"

Line 275—Author is James Russell III.

We thank the author for drawing our attention to this typo and fixed it.

Finally, we thank the reviewer again for taking the time to revise our manuscript and for providing insightful comments.